# Bicontinuous Gyroid Phase of a Water-Swollen Wedge-Shaped Amphiphile: Studies with In-Situ Grazing-Incidence X-ray Scattering and Atomic Force Microscopy

**DOI:** 10.3390/ma14112892

**Published:** 2021-05-28

**Authors:** Kseniia N. Grafskaia, Azaliia F. Akhkiamova, Dmitry V. Vashurkin, Denis S. Kotlyarskiy, Diego Pontoni, Denis V. Anokhin, Xiaomin Zhu, Dimitri A. Ivanov

**Affiliations:** 1Moscow Institute of Physics and Technology, Instituskiy per. 9, 141700 Dolgoprudny, Russia; grafskayaxeniya@gmail.com (K.N.G.); deniano@yahoo.com (D.V.A.); 2Institute of Problems of Chemical Physics, Russian Academy of Sciences, Chernogolovka, 142432 Moscow, Russia; azigy@mail.ru (A.F.A.); dmvav21@gmail.com (D.V.V.); 9920860@gmail.com (D.S.K.); 3Partnership for Soft Condensed Matter (PSCM), ESRF–The European Synchrotron, 71 Avenue des Martyrs, 38043 Grenoble, France; diego.pontoni@esrf.fr; 4Faculty of Chemistry, Lomonosov Moscow State University (MSU), GSP-1, 1-3 Leninskiye Gory, 119991 Moscow, Russia; 5DWI–Leibniz-Institute for Interactive Materials e.V., D-52056 Aachen, Germany; zhu@dwi.rwth-aachen.de; 6Institute for Technical and Macromolecular Chemistry of RWTH Aachen University, Forkenbeckstr. 50, D-52056 Aachen, Germany; 7Institut de Sciences des Matériaux de Mulhouse-IS2M, CNRS UMR 7361, Jean Starcky 15, F-68057 Mulhouse, France

**Keywords:** wedge-shaped amphiphile, double gyroid phase, grazing-incidence X-ray scattering, environmental atomic force microscopy, vapor annealing

## Abstract

We report on formation of a bicontinuous double gyroid phase by a wedge-shaped amphiphilic mesogen, pyridinium 4′-[3″,4″,5″-tris-(octyloxy)benzoyloxy]azobenzene-4-sulfonate. It is found that this compound can self-organize in zeolite-like structures adaptive to environmental conditions (e.g., temperature, humidity, solvent vapors). Depending on the type of the phase, the structure contains 1D, 2D, or 3D networks of nanometer-sized ion channels. Of particular interest are bicontinuous phases, such as the double gyroid phase, as they hold promise for applications in separation and energy. Specially designed environmental cells compatible with grazing-incidence X-ray scattering and atomic force microscopy enable simultaneous measurements of structural parameters/morphology during vapor-annealing treatment at different temperatures. Such in-situ approach allows finding the environmental conditions at which the double gyroid phase can be formed and provide insights on the supramolecular structure of thin films at different spatial levels.

## 1. Introduction

Molecular self-assembly is a spontaneous process during which the constitutive elements self-organize under action of non-covalent bonding forces [1,2,3]. Among self-assembling species, amphiphilic low-molecular-weight mesogens have been extensively studied because of their fascinating phase behavior [4,5]. In particular, wedge-shaped amphiphiles can self-assemble into a remarkable range of lyotropic liquid crystalline (LLC) and thermotropic liquid crystalline (TLC) mesophases with intricate one-, two-, and three-dimensional (1D, 2D, and 3D) periodic nanostructures [6,7]. Due to the unique structures of non-lamellar LC mesophases, they have attracted significant interest over the past few decades particularly for their applications in such areas as drug delivery [8,9], membrane protein crystallization [10,11], energy conversion and storage, gas storage, chemical sensing, and others [12,13].

Nanoporous membranes are already widely used in various applications, like chemical separation and purification, fuel conversion, and ecology [14]. They are perspective not only for classical chemical production but also for design of new sensors, electronics, medicine, and, especially, for fuel cells [15].

The most desirable structures for nanoporous materials are columnar or cubic phases. The use of columnar phases is more developed for conducting proposes but the control of channels orientation in such phases can be an issue. Thus, for optimizing the ion transport across the membrane in the columnar phase, one has to induce vertical or, the so-called homeotropic, orientation of the columns [16]. This is a nontrivial task because, in many instances, such orientation is not thermodynamically stable. The bicontinuous cubic phases do not require macroscopic orientation because the channels in such structures are running along all three directions in space. The cubic phases can be described in terms of triply periodic minimal surfaces (TPMSs). The TPMSs represent nonintersecting surfaces with three-dimensional (3D) periodicity and vanishing mean curvature H, where $H=\frac{1}{2}\left(\kappa_{1}+\kappa_{2}\right)$, and $\kappa_{1}$ and $\kappa_{2}$ stand for the principal curvatures at the point [17]. Such continuous surface separates the space into two interwoven nets of channels known as bicontinuous structure. The most commonly known TPMSs are the Schwarz primitive (P), Schwarz diamond (D), and Schoen gyroid (G) surface structures.

The TPMSs are intriguing due to their ability to exhibit unique physical properties. For example, such surfaces are used to model different crystalline structures in both natural and synthetic systems. Examples of these structures include LLCs and TLCs [18,19], block copolymer (BCP) self-assemblies [2,20,21,22] and organic zeolites [23]. It is noteworthy that one important structural difference between the LC and BCP systems consists in the values of the unit cell parameter, which ranges between ca. 10 and 100 nm for BCP-based systems [24], while it can rarely exceed several nanometers for LC systems [25]. In some instances, it is interesting to make the characteristic distances of both systems closer to each other, i.e., by increasing the sizes of the LC systems and/or by bringing down the dimensions of phases generated by the BCPs in order to make them interesting for the developing areas of nanotechnologies and nanopatterning [26].

The wedge-shaped molecules can be considered as building units for design of zeolite-like materials that can be excellent candidates for fabrication of nanoporous membranes [27,28,29,30]. It is already established that these compounds are able to self-organize in complex morphologies pertinent to the cubic bicontinuous phases, such as diamond, gyroid, and primitive cubic phases. The family of bicontinuous cubic phases can be considered as an organic equivalent of zeolites due to their developed network of well-defined nanochannels. Among these structures, gyroid bicontinuous phases are considered as being more accessible for practical applications because of self-supporting frameworks with better mechanical strength, ensuring open-cell character, high and uniform porosity, and large and predictable specific surface area. However, the gyroid phases formation mechanism and corresponding functionality are still not sufficiently studied. This is in part accounted for by experimental difficulties in studying the structure formation process in different environments with the necessity to perform in-situ structural characterizations. Therefore, several scientific and technical challenges have still to be taken to get a full understanding of the corresponding structure-property relationships, as well as details of the gyroid phase formation process.

For several years, our group has been involved in studies of self-assembly processes in various wedge-shaped compounds based on sulfonic acid. We have previously shown that, during heating or UV-irradiation (for the systems with light-sensitive groups), these systems form a variety of LC morphologies, such as smectic, columnar, and cubic (double diamond and gyroid). Swelling of thin films in solvent vapors in some cases leads to development of ion channel networks, with the lattice parameter becoming comparable to the one in the BCP systems [31,32,33,34,35]. Earlier, we have reported on formation of well-organized cubic phase in the films of wedge-shaped salts with linear alkyl chains during swelling in methanol. Surprisingly, this mesophase was found to be rather stable in a wide temperature range, and, upon preparation, the methanol solvent can be replaced by water in ion channels, which makes these systems interesting for future applications [33,34,35]. Synchrotron Grazing-Incidence X-ray scattering (GISAXS) experiments in combination with computer simulation reveal development of water channels of ca. 2 nm in diameter (cf. Figure 1), which makes them a promising candidate for fabrication of mechanically stable membranes with excellent proton conductivity.

In the present study, we address the development of ion-channel network during swelling in humid atmosphere by in-situ monitoring the structural evolution and topography of thin films of a wedge-shaped amphiphile.

## 2. Materials and Methods

The synthesis of pyridinium salt of 4′-[3″,4″,5″-tris-(octyloxy)benzoyloxy]azobenzene-4-sulfonic acid (C8AzoPyr) was described in previous reports [31,36]. The chemical structure of C8AzoPyr is given in Scheme 1.

Differential Scanning Calorimetry (DSC) measurements were carried out using a Netzsch DSC 204 unit. Samples (typical weight of 5 mg) were enclosed in standard Netzsch 25 µL aluminum crucibles. Heating and cooling rate was 10 °C/min.

Grazing-Incidence Small- and Wide-Angle X-ray scattering (GISAXS-GIWAXS) measurements were performed at the BM26 and ID10 beamlines of the European Synchrotron Radiation Facility (ESRF) in Grenoble (France) using a custom-designed environmental chamber [34,35].

The energy of X-ray photons was 12 keV. The s-axis (|**s**| = 2sin θ/λ, where θ is the Bragg angle, λ is the wavelength, and |**s**| is the norm of the **s**-vector) was calibrated using several diffraction orders of silver behenate. X-ray patterns were recorded using a 2D Pilatus 1M camera. The X-ray data analysis, including background subtraction and radial integrations of the 2D patterns were accomplished using home-built routines designed within the IgorPro software package (Version 6.37, Wavemetrics Ltd., Portland, OR, USA). For the GISAXS experiments, thin films of C8AzoPyr were prepared from a chloroform solution (20 mg/mL) by spin-coating (500 rpm/min) on a silicon wafer substrate. The phase composition of the thin films at different temperatures was addressed by in-situ heating and cooling of the samples in temperature range from –50° to 100 °C. The change in the phase structure during the water uptake process in thin films was also monitored using in-situ control of relative humidity (RH) and temperature.

A specially designed environmental cell was used for in-situ studies of the structural evolution in thin films (Figure 2). The compactness of experimental cell makes it possible to use it on the synchrotron beamlines, i.e., in the sample chambers of a diffractometer or spectrophotometer, and can be also combined with an optical microscope. The experimental cell allows one to study thin films by the GIWAXS/GISAXS methods with simultaneous control of external factors, such as UV radiation, temperature, and atmosphere of solvent vapors. The experimental cell includes the following components: a sealed chamber, consisting of a heating element, a lid to create the desired atmosphere over the sample, a frame for mounting, and a temperature and humidity control system.

AFM imaging was performed in Tapping Mode using a Cypher S Asylum Research Atomic Force Microscope. In the experiments, Oxford Instruments AC240TS medium soft silicon cantilevers (Abingdon, UK) with a resonance frequency 67 kHz and spring constant 1.82 N/m were used. The spring constant of the cantilevers was measured by the thermal noise method. To conduct measurements at variable humidity, a special portable air humidity control system was employed. The system shown on Scheme 2a consists of a computer connected to a controller and aeraulic circuit of humidity regulator (Scheme 2b). The main controller regulates the flow of the compressed air through the system. In order to create humidity-controlled air, dry and humid air are mixed in the desired proportions with the help of Mass Flow Controllers (MFC) for different flow ranges connected to the main controller. This air is then redirected through the Atomic Force Microscope Cypher S Asylum Research equipped with humidity sensor that allows to maintain the required humidity value during the experiment. The AFM images were analyzed using the open source software Gwyddion (Version 2.58).

## 3. Results

The thermal behavior of C8AzoPyr was assessed by DSC. The DSC trace corresponding to the first heating exhibits two endothermic peaks (Figure 3, black line). The first peak with onset at 72 °C, and enthalpy 0.7 J/g is associated with a solid-solid state transition; the second intense endothermic peak with onset at 96 °C, and enthalpy 26 J/g can be attributed to melting of a crystalline structure. Interestingly, the second heating curve does not exhibit endothermic peaks anymore (Figure 3, red line). In our earlier report, crystallization of C8AzoPyr from a LC phase during annealing at room temperature was explained by a process of local ordering of linear octyl chains [35]. Appendix A shows FTIR spectrum of C8AzoPyr after a long storage at room temperature. In the magnified region from 3000 to 2800 cm^−1^, one can see that conformation-sensitive anti-symmetric and symmetric stretching vibration modes of CH_2_ group are positioned at 2921 and 2851 cm^−1^, respectively (cf. Appendix A). These values are close to those characteristic of all-trans conformation of n-alkanes–2920 and 2850 cm^−1^, respectively [37]. Consequently, in the crystalline state, C8AzoPyr possesses ordered octyl chains in a quasi-extended conformation. Above the melting point, the structure reorganizes to a columnar hexagonal LC phase with disordered alkyl chains (Appendix A). Since the second heating performed after 3 min at room temperature does not reveal any transitions, the LC-to-crystal phase transition at room temperature is likely to be a slow process (Appendix A). This gives one the time to manipulate the structure while in the mobile LC state by varying the external factors. In contrast, in the crystalline state, a rigid framework of side chains prevents any structural evolution in the atmosphere of water vapors or in vapors of organic solvents [35].

Figure 4a displays GISAXS pattern after long storage at room temperature. The ordered phase was indexed to a columnar monoclinic unit cell (Col_mon_), which has the following parameters: a = 58.9 Å, b = 50.0 Å, γ = 61°. However, in the LC state with liquid-like side chains which is generated upon cooling from 100 °C, the structure of C8AzoPyr is more sensitive to vapors of polar solvents, such as water or alcohols, due to their more effective diffusion through the amorphous alkyl periphery of the molecular wedges. Thus, swelling in methanol atmosphere for 2 h results in formation of a layer-like lamellar structure with parameter a_lam_ = 51.2 Å (Figure 4b). The corresponding indexed 1D-reduced diffractograms are presented in Appendix A.

Earlier, we demonstrated that during slow cooling from 100 °C in a methanol atmosphere the high-temperature columnar hexagonal structure transforms to a stable cubic double gyroid structure [35]. It is noteworthy that the normal to the film surface becomes parallel to the 211 reciprocal space direction, as was previously noted for another wedge-shaped mesogen [31]. After several months annealing at ambient conditions, the gyroid structure not only persists but even shows a definite improvement of its organization. POM images of the dry film (not shown here) confirm the absence of birefringence which is typical of cubic phases. The diffraction pattern of C8AzoPyr thin film demonstrates a set of peaks with d-spacings ratio of $\sqrt{6}:\sqrt{8}$, which is characteristic of the Cub_gyr_ phase (symmetry $Ia\bar{3}d$). The corresponding lattice parameter a_gyr_ is 117 Å (Figure 4c). The summary of the extinction rules for the different bicontinuous phases can be found in ref. [38].

The presence of bicontinuous network of opened hydrophilic channels makes the film capable of efficiently swelling in a humid atmosphere. A 2D GISAXS pattern of the thin film with the Cub_gyr_ phase corresponding to the sample kept for two days in saturated water vapors reveals a significant increase of the lattice parameter a_gyr_ to 138 Å. This is an indication of formation of bicontinuous networks of water nanochannels (Figure 4d). The average radii of water channels in gyroid phase in the as-prepared and swollen film is found to be 8 ± 1 and 10 ± 2 Å, respectively [35]. Consequently, formation of Cub_gyr_ phase stabilized by ordered alkyl chains demonstrates the outstanding ability of the system to absorb water without destruction of supramolecular organization with long-range order. The swelling process of C8AzoPyr was further studied by in-situ AFM.

Figure 5a shows a topographic image of a C8AzoPyr thin film exposed to relative humidity of 42%. The film clearly exhibits a terrace-like structure that can be quantitatively analyzed using height cross-sections, as shown in Figure 5b. In this case, the cross-section traced along the black solid line in Figure 5a reveals terraces with an average height of 47.5Å. Since the vertical direction of the film corresponds to the 211 reciprocal space vector of the double gyroid phase according to GISAXS, it is logical to assume that the corresponding step height should give the reticular distance of the (211) planes of $\frac{a}{\sqrt{6}}$. This allows computing the lattice parameter a_gyr_ of 116 Å.

However, if one wants to analyze the gyroid structure with more scrutiny, it would be necessary to image the film surface with higher resolution. An example of such measurement is given in Figure 6, where AFM phase images of C8Pyr thin films at different RH values are shown. Despite some noise present in the images (the images after filtering are shown in the insets), one can recognize the unique pattern of the 211 plane of the double gyroid phase (see, e.g., Reference [39]). The observed morphological features of the 211 plane can be viewed relative to the $1\bar{1}\bar{1}$ and $01\bar{1}$ reciprocal space vectors that direct along and perpendicular to the characteristic “knitting” features of the plane, respectively. The analysis in the Fourier space given in the right column of Figure 6 allows visualizing the characteristic in-plane periodicities. Thus, the image taken in the dry (RH = 20%) state of the film reveals the fundamental periodicity along the $01\bar{1}$ direction of 0.057 nm^−1^, which is equivalent to the distance of 17.5 nm in direct space. The periodicity in the vertical direction of 0.09 nm^−1^ provides the corresponding direct-space repeat of 11.1 nm. Taking into account the expressions for both distances given in ref. [39], one obtains the lattice parameter of the gyroid phase to be 12.4 and 12.8 nm, respectively. These values agree relatively well with the corresponding SAXS unit cell parameter, which validates such AFM-based analysis. A small difference (of ca. 3%) in the values computed for the horizontal and vertical directions can be accounted for by a small drift of the piezo along the slow scan direction.

The described reciprocal-space analysis makes it possible to quantitatively compare the AFM phase images of the gyroid phase taken at different values of RH. Indeed, the two images in Figure 6 can be confronted based on the positions of the characteristic peaks in the Fourier space. The computation of the unit cell parameter for the image taken at 100% RH gives the value of 13.5 nm, which is noticeably larger than the corresponding value in the dry state and is also close to the SAXS unit cell parameter measured for the long-term annealing of the sample in the saturated water vapors. The presented AFM results confirm the unusually high stability of the Cub_gyr_ phase in swelling-drying cycles.

## 4. Discussion

The present work provides an example of a wedge-shaped C8AzoPyr amphiphile bearing promise for applications in ion-selective membranes for separation, catalysis, and energy conversion. To optimize the ion transport, one has to prepare the C8AzoPyr membranes in a bicontinuous cubic phase, such as the double gyroid $Ia\bar{3}d$. The experiments reveal a very rich polymorphic behavior of C8AzoPyr both in LLC and TLC phases, which makes any search for the desired cubic phase exclusively with ex-situ techniques inefficient. Here, we show that such search employing a combination of environmental in-situ GISAXS-GIWAXS and AFM experiments is much more productive as one can easily operate the external factors, such as solvent vapors and temperature, while continuously monitoring the film structure.

An interesting particularity of the studied system is that the gyroid phase can be locked in the film and persist even in its fully dry state. This is completely unusual for the LLC phases for which the gyroid phase exists only in a certain range of solvent fractions. Such locking the gyroid phase is accounted for by ordering of the octyl side chains that keep the gyroid phase stable in the course of repeated swelling-drying cycles. Such metastable state can be erased only if one melts the side-chains’ locks at high temperature. However, as long as the side chains keep their local ordering, the formed gyroid phase will persist, and the material will preserve its functionality. Therefore, it is shown that, by combining in-situ structure monitoring with reciprocal- and direct-space experimental techniques and optimized molecular architecture, one can design the materials with enhanced stability of the bicontinuous structure.

## 5. Conclusions

Using environmental in-situ grazing-incidence X-ray diffraction and atomic force microscopy, the phase behavior of thin films of a wedge-shaped mesogen in dry and swollen state was investigated. The studied samples show the presence of unusually stable bicontinuous cubic double gyroid phase, whereas, in the swollen thin films, they reveal formation of swollen water channels. The obtained results can help understanding the structure formation in supramolecular systems at different hierarchical levels and, consequently, contribute to the development of new approaches in fabrication of self-organized films with the desired morphology.

## Data Availability

Data available on request due to restrictions e.g., privacy or ethical. The data presented in this study are available on request from the corresponding author. The data are not publicly available due to organization policy.

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
