# Peer review of "Bicontinuous Gyroid Phase of a Water-Swollen Wedge-Shaped Amphiphile: Studies with In-Situ Grazing-Incidence X-ray Scattering and Atomic Force Microscopy"

_materials, 2021, doi:10.3390/ma14112892_

Round 1

Reviewer 1 Report

In the manuscript, the authors used environmental in-situ grazing-incidence X-ray diffraction and atomic force microscopy to investigated the phase behavior of thin films of a wedge-shaped mesogen in dry and swollen state. The presence of unusually stable bicontinuous cubic double gyroid phase, whereas in the swollen thin films they reveal formation of swollen water channels.

It is interesting results. However, in order to better clarify the phase behavior of thin films, the manuscript is recommended for publication after minor revisions.

  1. The average radius of water channels should be evaluated and give more discussion.
  2. X-ray diffraction (XRD) patterns at 80 and120oC of temperatures and 1st and 2 nd heating were provided to confirm the phase behaviors in detail.
  3. IR data should be added to present insights on the state of alkyl tails bands.
  4. If possible, Raman of samples should be provided.

Author Response

We would like first of all to thank the reviewer for the careful reading of our manuscript. Here are our answers on the reviewers’ questions/comments:

  1. The average radius of water channels should be evaluated and give more discussion.

The radii of water channels are now explicitly given in the main text of the revised article.

We added a new sentence on lines 228-230: “The average radii of water channels in gyroid phase in the as-prepared and swollen film is found to be 8±1 and 10±2 Å, respectively.[36]”

A detailed discussion on the calculation of the water channel radius is given in our previous work #36: Grafskaia, K.N.; Anokhin, D.V.; Zimka, B.I.; Izdelieva, I.A.; Zhu, X.; Ivanov, D.A. An “on–off” switchable cubic phase with exceptional thermal stability and water sorption capacity. Chem. Comm. 2017, 53, 13217-13220.

  1. X-ray diffraction (XRD) patterns at 80 and120oC of temperatures and 1st and 2 nd heating were provided to confirm the phase behaviors in detail.

The requested 2D and 1D-reduced GISAXS patterns recorded during first and second heatings of dry C8AzoPyr films at 80 °C and 100 °C  are now added to Supplementary Materials (Figs. S2 and S3). The phase behavior of the films is described in lines 190-194 of the main text: “Above the melting point, the structure reorganizes to a columnar hexagonal LC phase with disordered alkyl chains (Figure S2a,b). Since the second heating performed after 3 min at room temperature does not reveal any transitions, the LC-to-crystal phase transition at room temperature is likely to be a slow process (Figure S2c,d).”

  1. IR data should be added to present insights on the state of alkyl tails bands.

According to the reviewer’s suggestion, an IR spectrum of C8AzoPyr powder after long-term storage at room temperature is shown in Fig. S1. A comment on the state of alkyl chains is added to the main text (Lines 185-189): “Figure S1a shows FTIR spectrum of C8AzoPyr after a long storage at room temperature. In the magnified region from 3000 to 2800 cm-1 one can see that conformation-sensitive anti-symmetric and symmetric stretching vibration modes of CH2 group are positioned at 2921 and 2851 cm-1, respectively (cf. Fig.S1b). These values are close to those characteristic of all-trans conformation of n-alkanes – 2920 and 2850 cm-1, respectively.[38]”

A new reference #38 is accordingly added to the article:

  1. Snyder, R. G.; Strauss, H.L.; Elliger, C.A. Carbon-hydrogen Stretching Modes and the Structure of n-Alkyl Chains. 1. Long Disordered Chains. J. Phys. Chem. 1982, 86, 5145 —5150.

  1. If possible, Raman of samples should be provided.

Unfortunately, we were unable to record a Raman spectrum of the studied compound because of intense fluorescent background due to the presence of aromatic moieties. However, we believe that the evidence of the alkyl chains aggregation gathered both from X-ray scattering and IR spectroscopy is sufficient for our purpose. We will take into account the reviewer’s suggestion in our future work on similar compounds.

Reviewer 2 Report

This manuscript presents an interesting study of a wedge-shaped C8AzoPyr amphiphile. It is a very important topic in the field of stabilizing desired morphology of self-organized films. The contents are well organized for the readers to easily understand the most important issues in this field. Thus, I recommend the paper is published in the Materials after minor revision as follows.

1)  Can the authors show the 1D integration traced extracted from 2D XRD patterns (Figure 4) as supporting information?

2) In Figure 2 (b), can the authors specify the components of the cylinder and box next to number 5, respectively?

Author Response

We acknowledge the reviewer for the careful reading of the manuscript and valuable remarks/suggestions. Here are our answers :

  • Can the authors show the 1D integration traced extracted from 2D XRD patterns (Figure 4) as supporting information?

The 1D-reduced diffractograms corresponding to 2D patterns in Figure 4 of the manuscript are now added to Supplementary Materials (Fig. S3). The indices of the main reflexes and their assignment to the different phases are shown on the curves.

2) In Figure 2 (b), can the authors specify the components of the cylinder and box next to number 5, respectively?

The components of the temperature and humidity control system are now indicated in the figure. The cylinder stands for a liquid nitrogen dewar and the control panel/computer is displayed as a box.

English language and style are fine/minor spell check required.

A certain number of style corrections has been made in the text of the manuscript. These minor changes are highlighted in yellow.